# The Relationship between “Zero Waste” and Food: Insights from Social Media Trends

**DOI:** 10.3390/foods12173280

**Published:** 2023-09-01

**Authors:** Daniela Šálková, Olesya Maierová, Lucie Kvasničková Stanislavská, Ladislav Pilař

**Affiliations:** 1Department of Trade and Finance, Faculty of Economics and Management, Czech University of Life Sciences Prague, 165 21 Prague, Czech Republic; salkova@pef.czu.cz (D.Š.); maierovao@pef.czu.cz (O.M.); 2Department of Management and Marketing, Faculty of Economics and Management, Czech University of Life Sciences Prague, 165 21 Prague, Czech Republic; kvasnickova@pef.czu.cz

**Keywords:** zero waste, food, Twitter, social media, lifestyle, leftover, climate change

## Abstract

Zero waste (ZW), the concept of reducing waste production, is now becoming a lifestyle trend. Social media is strengthening this by popularizing the movement and connecting related communities. ZW and food are closely related, since food waste is a significant sustainability issue; however, the exact relationship between ZW and food communication on social networks is not clear. This study analyzed user communication on the social networking site Twitter between July 2008 and April 2023 to determine how members communicated and shared topics related to ZW and food; an analysis of hashtag frequency was also conducted. During the study period, a total of 50,650 tweets with both #zerowaste and #food hashtags were recorded, written by 21,271 unique users from all over the world. Topic analysis identified the nine related topics: ZW lifestyle, leftover recipes, ZW events, food rescue, climate change, packaging, ZW stores, composting, and ZW restaurants; visual analysis indicated that these topics were closely connected, suggesting common membership of these communities. Overall, our results provide insight into the ZW and food communities on Twitter, which may be useful for marketers, influencers, and government agencies to create targeted content and facilitate wider adoption of a ZW lifestyle.

## 1. Introduction

Empirical research on social media communication around zero waste (ZW) principles and food is limited. Previous studies have predominantly focused on sustainable lifestyle practices and climate change activism through everyday routines, most commonly on Facebook and Instagram. Most of the research used in-depth, semi-structured interviews with selected social media users as a method of data collection [1,2,3,4]. However, a more detailed analysis of ZW communication relating to food has not been conducted, nor has there been a focus on related topics, e.g., packaging and packaging materials [5]. Refs. [6,7,8] Twitter is the most popular and active place for food- and environment-oriented communities, compared with, for example, Facebook or YouTube [9]. Hashtags as a tool for sharing certain topic areas are very popular: almost 40% (39.2%) of all tweets contain at least one. Since approximately 500 million tweets are sent per day, almost 200 million tweets per day contain a hashtag [10]. Understanding communication on social media is essential to understanding the values, attitudes, experience, and behavior of people interested in ZW and food [7]. Currently, this topic is very popular and, therefore, there are a number of potential applications of the collected information; for example, identifying areas of related communication, including sharing related topics, can be used for marketing towards certain consumer groups.

The aim of this paper was to identify the key topics and hashtags associated with the hashtags #zerowaste and food on the social network Twitter. We focus on the broader context of content discussed in online communities in the context of ZW and food using social network analysis and social media analysis methods using the SMAHR (social media analysis based on hashtag research) framework, which has already been used to analyze themes related to organic food [6], healthy food [7] and cultivated meat [8].

### 1.1. Theoretical Background

The topic of ZW belongs to alternative approaches to waste treatment in relation to the circular economy [11] and is becoming increasingly popular. Marciano (2021) states that the basic principles and conditions for effective ZW strategies include the complete removal and recycling of waste [12]. The need to work with ZW principles is not only related to the ethical aspect of waste, but also to the issue of global pollution. According to Lee et al., waste is mostly incinerated or goes to landfill, which, apart from being an unsatisfactory form of waste management, is also a drain on societal resources [13]. The implementation of ZW in some countries is preventive and is based on ethical considerations. In other countries, where increasing food waste poses a real threat, the government has intervened in the form of regulations controlling waste treatment and management. For example, in 2018, China introduced a plan to implement a ZW strategy in cities [14]; in other countries, such as France, waste recovery options are addressed by specialized agencies [15].

Food sustainability is a key issue in sustainability and waste production. When modeling variations of waste management strategies using the example of social events, including ZW, one of the main problems is the difficulty in predicting food demand and influencing consumer behavior [16]. Approximately one third of food for consumption is wasted or thrown away in households globally [17]. Promoting methods to reduce food waste [18] as an important element in reducing waste production and is therefore an important area of ZW; food waste recovery through advanced technologies is one of the ways to do this [19,20,21,22]. There is a close link between the concept of ZW and food, as evidenced by the United Nations’ goal of “zero hunger” [23]. More than 80% of food is compromised during the phases of production, processing, and consumption [24,25], and food waste is responsible for a fifth of all agricultural water consumption and an annual economic loss in the amount of USD 940 billion [26]. Practices to reduce food and packaging waste through the self-production of food and by preparing food from scratch are other ways to facilitate food waste reduction from households and individuals [15,27].

Social networks play an important role in today’s communication and are becoming a fixed part of people’s daily lives, not only in adults but also in adolescents [28], who are more likely to look to social networks for support in substitution of real-life communication [29]. Social media also has a significant impact on people’s behavior [29]. Nelson et al. (2019) reported that social networking is related to lifestyle preferences and even dietary patterns, especially in women [30]. In addition, social networks are an effective tool for promoting and spreading certain topics among people. The ability of social networks to influence mental state [31] and thoughts, and to promote trends in a large sample of users, makes them an effective tool for promoting ZW, especially since Twitter is often used as an educational resource [32]. The social network Twitter is one of the most popular platforms [33], meaning that it provides high-resolution data for an analysis of ZW; indeed, the potential of Twitter data for analyzing human behavior was previously mentioned by Vidal et al. [34]. However, despite social networking’s increasing integration into people’s lives, it is still relatively new and has not been fully explained or described in theoretical terms. For this reason, further analysis focusing on specific contexts (e.g., ZW and food) and the connections between topics and communities is required.

#### The Importance of Social Network Analysis in Zero Waste

Social media enables information sharing and social interaction within a group or groups of people [35]. There is no clear definition of the term social media; however, a common feature of social media is that it allows users to communicate, collaborate, create, edit, and share content [7]. Social media can serve as a tool to combat negative stereotypes and to build social capital and image [36]; this also applies to topics such as ZW and food.

Social networks are commonly used in marketing. In the study by Chodak et al., they showed that different effects were obtained depending on the content of posts (i.e., image or video) related to the same product. While video achieved the highest level of reach, images achieved the highest level of engagement [37]. This is used by some catering facilities when they work with the concepts of ZW and recycling in advertising campaigns. As interest in the issue of waste grows, some consumers are choosing businesses whose waste management aligns with their preferences. Active promotion of ZW and food waste reduction has therefore raised interest in real life as well as on social media.

As waste is one of the most serious environmental issues today, various means are being used to raise awareness of the importance of this issue among the general public. For example, before reading posts related to sustainability and waste issues, most users consume products in large quantities and mostly in plastic packaging, but this behavior changes after engaging with waste-related posts [38]. Therefore, social media increases attempts to avoid plastic consumption as the reader’s interest in the topic increases, and as long as they are able to engage with the content, e.g., to read a link attached to a post, these conditions also affect the likelihood that the tweet is shared further. This study demonstrates that social media can influence consumer behavior around plastic reduction if the information comes from credible sources, can easily produce results, and has a direct impact on their own health [38]. Contrasting findings can be used as models to create social media posts that can influence consumer behavior around reducing waste pollution. However, user characteristics such as gender play a key role in the intention to reduce plastic use or to engage with the topic of ZW [39]. Other key factors, according to Twitter data, are the education and economic level of the person: poverty and low education level are negatively correlated with a willingness to engage with or to reduce the amount of plastic used [40].

Public understanding of the issue of waste is ambiguous [41]. While communication of this content on Twitter and other social media has grown over the last few years, the translation of ZW principles into the lives of individuals has lagged. Active promotion of the issue on social media may remain confined to the virtual world, although some findings suggest that strengthening environmental self-identity is an effective way to encourage pro-environmental action, since people with a strong environmental self-identity are likely to act in an environmentally friendly manner even without external encouragement [42]. In addition, having a strong moral self-identity motivates prosocial behavior, since individuals seek to demonstrate their moral qualities to others [43]; for example, sharing content that has high moral value (i.e., relating to ZW and food) evokes a positive impression of those sharing this content. Studies show that social media analysis based on analyzing posts containing hashtags provides information for theoretical use in scientific research and practical application in multiple research areas [44]. A hashtag is a keyword by which a post is easily traceable. The first large-scale research on online ZW communities was conducted by Pedersen (2017), who used a digital ethnographic approach to study social media interactions in a ZW community in Denmark [45]. The purpose of social media analysis is to examine the content of communication among users, which includes an analysis of hashtag use and the intensity of interactions.

Climate change activism is an important topic that defines the space for discussion and proposals for action in society [46,47,48]. ZW is an important element for individuals who share climate change-related content [49], since preventing waste can preserve current resources and slow down waste-generation-associated climate change; this is now considered a matter of social responsibility [50].

## 2. Materials and Methods

The SMAHR framework of Pilař et al. (2021) was used to analyze Twitter communication [44]. The analysis included the following steps:
1.Data acquisition: Data were obtained using the Tractor 4.0 [51] software. Data were downloaded on 16 May 2023 for the period between 1 July 2008 and 30 April 2023 (to preserve the whole quarter). The query term for the data download was “#zerowaste” AND “food”. During this period, 50,650 tweets were generated from 21,271 unique users. The data were saved in CSV format.2.Data mining: Communication on the social network Twitter was analyzed using the following methods:(a)Frequency analysis: A basic characteristic where the most-communicated hashtags were identified [52].(b)Topic analysis: Topic analysis allows primary themes or subjects conveyed within an extensive dataset, such as social media posts, to be identified. Within intricate networks, such as social media networks, certain nodes (e.g., hashtags or words) exhibit a higher level of interconnectivity between themselves than with the remainder of the network; clusters of individual hashtags can be utilized to identify topics. The objective of this phase was to ascertain the topical framework of discussions pertaining to ZW and food on Twitter. Unlike frequency analysis, which relies solely on hashtags, topic analysis includes all text in tweets. The Graphext software (8A2023) [51] was employed to conduct the topic analysis, using a modified iteration of the Louvain algorithm [53]. The network was constructed based on interconnections between individual words within tweets. The Louvain algorithm utilizes an iterative process that assigns nodes to clusters, aiming to optimize a performance metric known as modularity. This metric evaluates the relative density of edges within clusters compared with those between clusters. The calculation for determining the number of distinct communities within the dataset is as follows:
(1)$$\Delta Q=\left[\frac{\sum_{in}^{}+2k_{i,in}}{2m}-{\left(\frac{\sum_{tot}^{}+k_{i}}{2m}\right)}^{2}\right]-\left[\frac{\;\;\sum_{in}^{}}{2m}-{\left(\frac{\sum_{tot}^{}}{2m}\right)}^{2}\right]$$ where ∑_in_ is the sum of weighted links inside the community, ∑_tot_ is the total number of weighted connections inside the community, k_i_ is the total number of weighted links related to community hashtags, k_i,in_ is the total weighted linkages from an individual to community hashtags, and m is the normalization factor, calculated as the total weighted links over the entire graph [53].(c)Visual analysis: The utilization of network visualization techniques, such as force-directed layouts, enables elucidation of various facets of a network, including interconnectedness density and topic polarization. The objective of this phase was to discern the polarity associated with the identified topics. A two-dimensional graph was generated to facilitate visual analysis using the ForceAtlas2 layout algorithm, which is an enhanced version of ForceAtlas designed for handling large-scale networks. This approach uses visual representations of smaller samples to identify within-community items [54]. Graphext software [51] was employed to create the visual analysis.3.Knowledge representation: Knowledge representation is a methodology that uses visualization tools to explicate the outcomes of data mining. It amalgamates individual variables and outputs derived from the data evaluation phase to highlight the significant findings of preceding analyses.

## 3. Results and Discussion

First, an analysis of hashtag frequency in published tweets relating to ZW in combination with the term “food” (see methodology) was conducted (Table 1). Other hashtags associated with #zerowaste represented a diverse spectrum of subtopics, suggesting that social media users aware of ZW had a broad range of interests.

Given the focus of the research, the hashtags #zerowaste, #foodwaste and #food appeared, as expected, in the top three places in the frequency analysis. In fourth and fifth place, the link between ZW and sustainability was identified with the hashtags #sustainability and #sustainable, respectively; these were closely followed by the hashtags #plasticfree, #ecofriendly, #recycle, and #recycling in the top ten.

Solid waste has become a global environmental issue [55,56] due to rapid economic and population growth [57], and the ZW movement aims to minimize the growing amount of this solid waste. However, according to a 2022 OECD study, only 9% of the world’s total plastic waste is recycled [58]. The results of our analysis confirm that Twitter users are aware of the relationship between the concept of ZW and plastic waste production and include this topic in their discussions by sharing the hashtag #plasticfree. Twitter users also associate #ecofriendly with the hashtags #zerowaste and #food. A number of papers [59,60,61] confirm the benefits of waste prevention for the environment. Therefore, improving environmental awareness and supporting ecological balance through the behavior of individuals is key to saving materials and energy and to supporting the principle of ZW [62]. The use of the hashtags #climatechange and #circulareconomy, which are also closely correlated with the concept of #zerowaste [63], suggest that individuals interested in ZW are aware of larger environmental and societal concerns. However, there is a significant discrepancy between the perception of ethical principles and the importance of the environment for future generations and the willingness to behave ecologically in everyday life. To transition to a circular economy, the involvement of all actors (including individuals), including increased responsibility and awareness, is required [64]. However, although approximately one third (30%) of consumers state that they are very interested in environmental issues and try to reflect this in their purchases, the share of ethical food on the market remains at only 5% of sales [65].

Refs. [63,64] The topic analysis identified the following nine largest topics: (1) ZW lifestyle, (2) leftover recipes, (3) ZW events, (4) food rescue, (5) climate change, (6) packaging, (7) ZW stores, (8) composting, and (9) ZW restaurants (see Table 2).

Twitter is becoming increasingly important for sharing information [11], where common topics are communicated between users on a larger scale in the form of thematically oriented communities. These so-called communities enable more targeted interactions and more relevant conversations among users [66]. We found that the largest topic associated with #zerowaste was “ZW lifestyle”. The popularity of this topic suggests that for social network users, ZW represents a way of life rather than a single activity, resulting in a series of subsequent activities [49] to reduce individual waste production. This topic focuses significantly on efforts to reduce waste in general but also encompasses the form of food waste. This topic was most strongly communicated among the Twitter users in terms of geographic distribution in the United States, the United Kingdom, and India.

The second-largest topic was “leftover recipes”, which was strongest in the United Kingdom. The users of this community often discussed the subtopic of veganism, suggesting a close connection between the concept of ZW and the preparation and consumption of food. The fifth-largest topic was “climate change”, confirming a close relationship between ZW and the impact of individual behavior on the planet. This group was geographically strongest in the United Kingdom, Ireland, and India.

The third-largest topic, “ZW events”, is devoted to events that are organized in connection with the concept of ZW. Within this topic, events to raise awareness of the concept of ZW and to allow like-minded people to get together were promoted. For example, “Hey Birmingham, what’re you doing on Jan 12th? Ahh yeah, that’s right. Nothing. Street food, wine & comedy in a zero-waste taproom? Sounds bloody cool eh?” and “We will be at Streets Alive! on September 18th at Streets Alive! festival hosted by @bikewalktomp. You are encouraged to BYO (Bring Your Own) for food and community dye bath #zerowaste”. As mentioned in a previous study [16], public events based on sustainable initiatives are on the rise. Zelenika et al. (2018) points out, however, that the success of such events depends on the support of volunteers [67].

The fourth-largest topic was “food rescue”, where instructions on how to minimize food waste in households were communicated.

The fifth-largest topic communicated the issue of “climate change” in connection with ZW. A number of studies point to the interconnectedness between food waste and climate change [68,69,70], since the main gases contributing to the greenhouse effect are closely related to food production and consumption [71].

The sixth-largest topic communicated on Twitter in connection with ZW and food was “packaging”, which deals with the plastic waste generated in connection with food.

Furthermore, the topic analysis identified topics dealing with the communication of “ZW stores” (seventh topic) and “ZW restaurants” (eighth topic). Tehrani et al. (2020) point out that the restaurant industry produces a variety of waste with a high potential for recycling, resale, reuse, and/or donation [72]. According to a previous study [73], reducing food waste is a key challenge for the food-service industry, but implementing measures to reduce food waste depends primarily on the beliefs, knowledge, and goals of restaurant management. However, many studies [74,75] prove that there is a dependency between green initiatives such as waste reduction and restaurant performance.

Figure 1 shows the development of the identified topics in the time period from July 2008 to April 2023. A general increase in topics related to the terms #zerowaste and “food” in communications on Twitter was observed in the period from January 2018 to January 2020. In the course of 2020, during the global COVID-19 pandemic, this communication decreased. A renewed interest in the identified themes was evident again at the start of 2022. Overall, during the entire period of COVID-19, a decrease in communication was recorded in all topics identified in our research, suggesting that even social networks reflected a shift in user interest from ZW toward health-related topics.

### 3.1. Visual Analysis

The visual analysis (Figure 2) showed that there were no polarized communities in this area, as evidenced by a single structure comprising all topics. This shows that individual topic communities connect and intertwine, that membership to multiple communities is likely, and that discussion around individual topics is communicated in one space.

### 3.2. Theoretical and Practical Implications

Based on the results of the thematic analysis related to ZW and food on Twitter, we identified several theoretical and practical implications.

#### 3.2.1. Theoretical Implications

This study provides insight into the role of social media, specifically Twitter, in spreading information and raising awareness about ZW and related topics. This research expands the current literature by examining the content of communications related to ZW and food, identifying the main topics and their development over a 15-year period. Results further show that there are a number of topics discussed on Twitter focusing on different aspects of ZW and food. Finally, analysis of the geographical distribution of topics offers insight into how topics and discussions spread across different countries and cultures. Therefore, this study contributes to the theoretical understanding of how ZW and food-related topics are interconnected, as well as the global nature of these communities.

#### 3.2.2. Practical Implications

Firstly, the results of our analysis can be used to create targeted content for those interested in ZW and related topics. One avenue of implementation would be marketing campaigns and strategies: marketers can use the keywords and hashtags listed in this study to create relevant content and engage their target audience. Our results may also help non-profit organizations, activists, and policy makers better understand how information about ZW and food spreads on social networks, which could improve their communication with the public and achieve a greater impact. Finally, the present results provide information for content creators and influencers dealing with ZW, food, and composting-related topics, who can use the results to target their content to the needs and interests of their audience, thereby increasing their reach and engagement.

In addition, by analyzing social media, the results of local government activity can be measured; for example, by assessing the extent of public interest in composting. Our results demonstrate that composting is one way people ameliorate food waste. Authorities could then consider this information, take advantage of the interest of local residents and offer them a solution that makes composting easier; this is particularly important in large urban agglomerations where composting options may be limited.

Twitter can also serve as an educational tool for users. The popularization and visibility of ZW can thus directly impact waste production and food waste at both the individual and community level. In particular, sharing information between social media users and presenting lifestyles that minimize the negative impact of consumption on the environment could directly lead, for example, to a reduction in the use of plastic packaging or to the management of food waste.

### 3.3. Future Studies and Limitations

Data research focused on the analysis of hashtags and keywords is associated with a number of methodological limitations. One of the most significant ones is that all analyzed tweets were written in English. The identified topics on Twitter reflect a relatively narrow geographic and linguistic scope, as most of the monitored groups are located primarily in English-speaking countries. The countries and territories with the largest audiences on Twitter in 2022 were the U.S., Japan, India, and Brazil [76]; therefore, these countries are likely to be over-represented. Sharing content through hashtags, which are usually in English, eliminates the language barrier to a certain extent, but language remains a limitation of the research.

Another limitation of this research is the age of Twitter users, which is not representative of the wider population. Younger and middle-aged people predominate, which is reflected in the method and content of communication. Specifically, 38.5% of Twitter users are between the ages of 25 and 34, and almost 21% are aged between 35 and 49 [77]. Nevertheless, these are users for whom environmental issues are important. Finally, it is not possible to identify the gender and ethnicity of users from the data downloaded from Twitter.

## 4. Conclusions

Waste has become a major global environmental problem [62]. The public is both the main producer of solid waste and the victim of environmental pollution [78]. The impact of human behavior on the environment is therefore important. Global consumers are increasingly interested in the negative effects of packaging waste on the environment and share their attitudes and experiences with each other, very often through social networks. A huge advantage of social networks compared with traditional communication channels (television, radio, newspapers) is their flexible two-way communication [79]. In addition, social networks are potentially powerful platforms to initiate behavioral change due to the large number of users. There are now 4.62 billion social media users worldwide, corresponding to more than 58% of the total world population [10]. Thanks to the power of social media, hashtags are becoming an effective tool to reach larger audiences and to attract attention using relevant hashtags. ZW is becoming a lifestyle trend that social media is reinforcing and helping to spread [80].

Social media has become a particularly important platform for discussing waste reduction in various contexts. The ubiquity of Twitter makes it a useful tool for encouraging discussion with conscious consumers who want to have dialogue on a given topic; therefore, Twitter can be helpful in promoting social behavior change. Social network analysis is gaining popularity in social media research because it can provide effective techniques to extract valuable information from massive amounts of network data [81]. Twitter communication and ZW communities are growing rapidly, but the deeper context of ZW activities through hashtags and their various connections have yet to be explored in detail. This article is one of the first to examine the structure of online communities engaged in ZW and food on Twitter, with the aim of better understanding the behavior of social network users and the mode and wider context of communication within ZW-related communities.

Overall, we found that ZW and food communities were closely related on Twitter, identifying nine topics that contained a high frequency of ZW and food-related content. These topics may be utilized in marketing communications, corporate social responsibility activities, and individual education to produce targeted content and to encourage wider adoption of a #zerowaste lifestyle.

## Figures and Tables

**Figure 1 foods-12-03280-f001:**
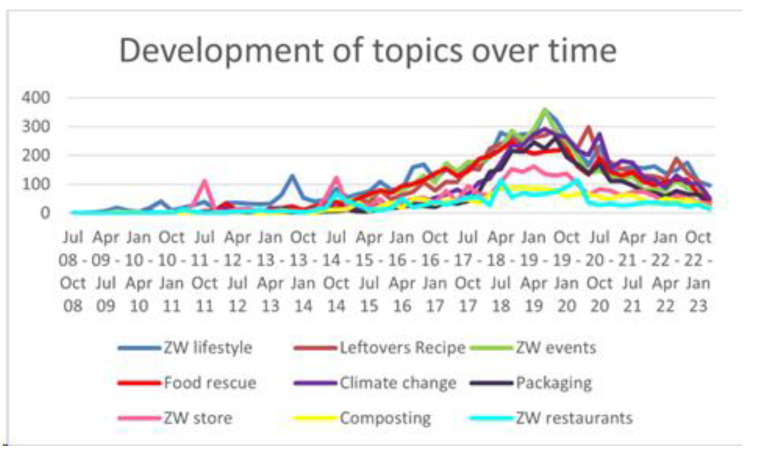
Development of zero waste and food-related topics on Twitter between July 2008 and January 2023.

**Figure 2 foods-12-03280-f002:**
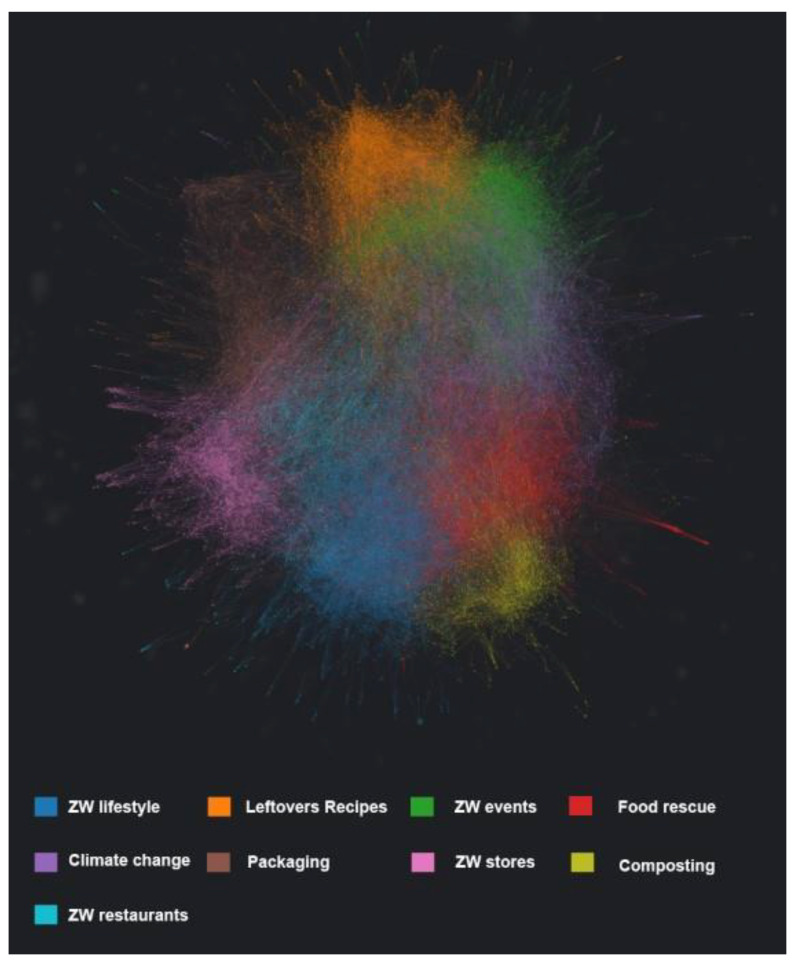
Visualization of zero waste and food-related topics on Twitter between July 2008 and January 2023. Individual points in the graph = individual tweets.

**Table 1 foods-12-03280-t001:** Most frequent hashtags from tweets in connection with the hashtag #zerowaste and the word “food” on the Twitter social network.

No.	Hashtag	Fr	No.	Hashtag	Fr
1	#zerowaste	33,250	21	#plastic	858
2	#foodwaste	10,836	22	#foodie	844
3	#food	4952	23	#composting	802
4	#sustainability	4024	24	#circulareconomy	762
5	#sustainable	2485	25	#sustainableliving	757
6	#plasticfree	2014	26	#giveaway	660
7	#ecofriendly	1864	27	#lovefoodhatewaste	633
8	#waste	1695	28	#gardening	618
9	#recycle	1446	29	#free	602
10	#recycling	1364	30	#community	589
11	#environment	1281	31	#savetheplanet	580
12	#compost	1260	32	#kitchen	544
13	#eco	1148	33	#homemade	540
14	#reuse	1132	34	#foodprep	538
15	#zerohunger	1050	35	#gogreen	536
16	#reduce	1033	36	#onthego	530
17	#vegan	976	37	#foodrescue	528
18	#organic	951	38	#app	526
19	#green	933	39	#kitchenessentials	522
20	#climatechange	872	40	#kitchengoals	519

Fr = Frequency of the hashtag.

**Table 2 foods-12-03280-t002:** Identified topics related to zero waste and food with country occurrence.

Number of Topics	Size of Topic	Name of Topic	Keywords	Largest Occurrence of Topic
1	18.43%	ZW lifestyle	zero waste; wasteweek; zerowasteweek; challenge; foodwaste; reduce; lifestyle; wastelifestyle; community; solution	United States, United Kingdom, India
2	14.71%	Leftover recipes	recipe; leftover; delicious; salad; vegan; soup; lunch; healthy; foodie; cook; meal; love; homemade; dinner	United Kingdom, Ireland, India
3	14.36%	ZW events	love; event; great; come; community; local; people; support; open; market	Ireland, Belgium
4	13.62%	Food rescue	food waste; reduce; tip; reduce food; sustainability; stop food waste; home; reducefoodwaste	India
5	12.48%	Climate change	#climate; climatechange; climateaction; sustainability; sustainable; nature; earth; future; circulareconomy; sustainable	United Kingdom, United States, India
6	9.05%	Packaging	plastic; bag; packaging; plasticfree; biodegradable; reusable; beeswax wraps; plasticpollution; use; recycle; bags; nonplastic	Canada
7	7.09%	ZW stores	waste grocery; store; zero waste shop; waste shop; supermarket; shopping; zero waste grocery store; food shopping, container	United Kingdom, Malaysia
8	4.95%	Composting	compost, composting, scrap, food scraps, landfill, soil, nyc, divert, garden, program, recycling, compostable	United States
9	4.54%	ZW restaurants	restaurant; waste restaurant; silo; chef; zero waste restaurant; zero waste bistro; bistro; cafe; bar; menu	United States

## Data Availability

All data used in this study can be downloaded via the Twitter API [82].

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
