# Peer review of "The Relationship between “Zero Waste” and Food: Insights from Social Media Trends"

_foods, 2023, doi:10.3390/foods12173280_

Round 1

Reviewer 1 Report

Review report - Foods 2532740 Manuscript

Summary of article

Aim of the paper:

The concept of Zero Waste entails the reduction of waste creation. In contemporary society, the adoption of Zero Waste practices is increasingly being observed as a popular lifestyle choice.

Social media platforms are playing a pivotal role in reinforcing this phenomenon and facilitating the dissemination of the movement among the wider populace. There exists a strong correlation between the concept of Zero Waste and the domain of food.

Food sustainability and waste management are critical concerns within the realm of sustainability and waste generation. This research examines user communication patterns on the social media platform Twitter spanning a duration of fifteen years.

The objective is to investigate the methods employed by users to communicate and exchange information pertaining to ZW, with a specific emphasis on the domain of food.

Simultaneously, a study was done to analyze the frequency of hashtags in relation to the concept of Zero Waste, specifically in conjunction with the topic of food.

Main contributions:

The following main contributions were observed with supporting results

The statistics were gathered throughout the period spanning from July 2008 to April 2023. In this particular timeframe, a comprehensive count of 50,650 tweets using the hashtags #zerowaste and #food was documented, originating from 21,271 distinct individuals spanning various geographical locations worldwide.

The topic analysis has found nine prominent subjects: Zero Waste (ZW) lifestyle, Leftovers Recipes, ZW events, Food rescue, Climate change, Packaging, ZW retailers, Composting, and ZW restaurants. The following are the subjects that establish a connection between lifestyle, namely waste reduction, and food management.

Strengths of article: Study methodology is designed in an elaborate and critical manner.

Comments on article

Title of article is quite long and shall be confusing for readers. Put a more concise and appropriate title.

Article is well written and needs major improvements throughout manuscript.

Figures and Tables needs to be represented clearly for sake of clarity to readers.

Appropriate Referencing needs to be improved in entire manuscript.

Specific comments

Title

Title of article is very lengthy and not clear.

Make a concise title.

Abstract

Revise abstract with clarity information related to - background, objectives, methods, results and conclusion with appropriate word count.

Introduction section

Introduction section is quite lengthy.

Combine all sections of introduction and make a focused introduction relating to concept of study.

Material and methods

Overall, improvise methodology section with elaborate details about study design, techniques employed.

Provide relevant details of statistical analysis employed to represent data.

Results and Discussions

Table 1 : Try to include most focused data by highlighting the key data points for the reader. Include table legends.

Table 2 : Try to include most focused data by highlighting the key data points for the reader. Include table legends.

Figure 1: Represent figure with more clarity and easily readable by readers. Include figure legends.

Figure 2 : Figure is confusing. Represent figure with more clarity and easily readable by readers. Include figure legends. Label figure appropriately.

Over all, Entire section needs to be revised with most focused text and tables to be represented in this section. Any supporting results can be included as supplementary information.

Conclusions

Conclusion section is repeating lots of information.

Rewrite conclusion clearly summarizing the main observations of study and key driving factors.

Include Recommendations for future research work

References

Please include information from latest reference articles as much as possible. 

Extensive english language required.

Author Response

First, we would like to thank you for the time you spent on the review. Your comments were valuable. We implemented them and overall moved our manuscript to a higher level.

We modified the article's title according to the comments and shortened the introduction. 

We have modified the material and methods to be more informative. We've highlighted essential points in Tables 1 and 2, and we've also edited Figures 1 and 2 and added legends. 

Overall, the article has undergone significant proofreading, and the conclusions have been modified to be more specific.

Current sources were also used and increased.

Thank you for your time and inspiring comments, which help us to improve the manuscript.

Now we hope that the manuscript meets all requirements for publication.

Reviewer 2 Report

The objective of the manuscript entitled: “Food in the context of zero waste: lifestyle, leftover cooking and climate change” was to identify the significant topics and hashtags that are connected to the hashtag #zerowaste and are used in conversations about food on the social media platform Twitter. The theoretical features of the research area on zero waste (ZW) as a waste reduction concept in the context of food and the significance of social networks in communication were discussed in the paper. In discussion the findings and their implications for practice and future study about the role of social networks, particularly Twitter, in communicating the subject of zero waste related to food and sustainability were elaborated. The downsides as well as restrictions of the study were also provided.

In the Introduction part of the manuscript, the subject was well addressed and citation of the literature is adequate and up to date.

The Material and Methods are well presented and the methods used are appropriate for this kind of research.

The Result and the Discussion part of the article tackle the subject of the research appropriately, and elaborate the findings well, by making appropriate conclusions and references to other previously conducted researches.

The References are up to date and refer to the subject correctly.

Author Response

We would like to thank you for the time you spent on the review.

We are glad that you like the article.

Reviewer 3 Report

Twitter data was used for this study. What about LinkedIn? Can we also use LinkedIn data for this study? It is also a social engagement platform with more users. 

As discussed in this paper, there is no information about the age, gender, or ethnicity of the people and Twitter network does not represent the composition of the world population. The age group of Twitter network is already give more importance to environmental issues, so how can we draw a conclusion that social media strengthening this trend? Can we compare different social platforms to eliminate age, gender etc. effect?

Author Response

We thank you for the time you spent on the review.

We are glad that you like the article.

Unfortunately, Twitter no longer provides this information, and after the Cambridge Analytic scandal, social networks such as Facebook and Instagram are already closed for downloading data.

We still hope it will be possible to use these platforms for purely academic purposes.